# A Newly Reported Parasitoid, *Pentatomophaga latifascia* (Diptera: Tachinidae), of Adult *Halyomorpha halys* in Beijing, China

**DOI:** 10.3390/insects11100666

**Published:** 2020-09-29

**Authors:** Juhong Chen, Wenjing Li, Qianqian Mi, Feng Zhang, Shusen Shi, Jinping Zhang

**Affiliations:** 1MARA-CABI Joint Laboratory for Bio-Safety, Institute of Plant Protection, Chinese Academy of Agricultural Sciences, No. 2 Yuanmingyuan West Road, Beijing 100193, China; 15181674153@163.com (J.C.); wenjingli321@163.com (W.L.); q.mi@cabi.org (Q.M.); F.zhang@cabi.org (F.Z.); 2College of Plant Protection, Jilin Agricultural University, No. 2888 Xincheng Street, Changchun 130118, China; sss-63@263.net

**Keywords:** *Halyomorpha halys*, adult, *Pentatomophaga latifascia*, parasitoid, parasitism, biological control

## Abstract

**Simple Summary:**

*Halyomorpha halys* (Hemiptera: Pentatomidae) is a well-known invasive pest that feeds on plant and fruit tissues. Despite numerous studies on egg parasitoids of *H. halys*, the natural enemies of the nymphs and adults remain poorly known. In this paper, we surveyed the parasitoids of adult *H. halys* by collecting overwintering *H. halys* populations. Our results showed that *Pentatomophaga latifascia* (Diptera: Tachinidae) had laid eggs on the surface of adult *H. halys*, and the hatched larvae of *P. latifascia* then penetrated the host body and fed internally to complete their development. The average parasitism rate of *P. latifascia* on *H. halys* was 2.42%. These results add an important piece of knowledge about the natural enemy community attacking *H. halys* in its native range and may have useful implications for biological control in the newly invaded areas.

**Abstract:**

*Halyomorpha halys* (Stål) (Hemiptera: Pentatomidae) is a serious pest in agriculture and forests, as both adults and nymphs feed by piercing the surface of the plant and fruit tissues, causing damage. The eggs of *H. halys* are commonly attacked by parasitoids, however, the nymph and the adult are rarely attacked by natural enemies. We surveyed the parasitoids of adult *H. halys* by collecting samples from overwintering populations at three different locations and checked their body surfaces for the presence of tachinid eggs. Any host adults carrying tachinid eggs were reared in a cage for further species identification. We found that the eggs of *Pentatomophaga latifascia* (Villeneuve) (Diptera: Tachinidae) were laid on the surface of *H. halys*, and the hatched larvae penetrated the host body and fed internally to develop. The last larval instar emerged from the host to develop into pupae, killing the host in the process. According to the field survey, the average parasitism of *H. halys* by *P. latifascia* was 2.42%. The parasitoids of adult *H. halys* in their native range have so far been little studied and may provide a complementary component of egg parasitoids for biological control against *H. halys* in invaded areas.

## 1. Introduction

The brown marmorated stink bug, *Halyomorpha halys* (Stål) (Hemiptera: Pentatomidae), is a serious pest native to China, Japan and South Korea. It was accidentally introduced into North America in the fall of 1996 [1], Europe in 2007 [2] and South America in 2017 [3]. *Halyomorpha halys* is still spreading across the world and has been repeatedly intercepted at the border in Australia and New Zealand [4]. *Halyomorpha halys* develops via incomplete metamorphosis, a process that involves three stages: eggs, nymphs and adults. The nymphs and adults of *H. halys* cause serious damage to agricultural and forestry plants by inserting their stylet into the fruits and stems of their hosts [5,6]. For example, 50–80% yield loss was caused by *H. halys* on peaches and pears in the 1980s in northern China [7,8]. More than 50% of early maturing pears were damaged and approximately 30% of kiwifruit was lost because of *H. halys* in Italy [9]. In the eastern United States, 25% of apples and peaches were damaged [10], and $37 million in losses were caused by severe outbreaks of *H. halys* in the apple-growing area of the Mid-Atlantic USA in 2010 [11]. Apart from being an agricultural pest, *H. halys* is considered a nuisance problem, as massive numbers of adults often invade human-made structures to overwinter inside protected environments in urban areas [12].

*Halyomorpha halys* has quickly spread throughout North America and Europe and become a serious pest in recent years. This is attributed to its strong flight dispersal capacity, the hitchhiking behavior of overwintering populations [13,14], and the absence of natural enemies in newly invaded areas [15]. In commercial agricultural settings, as well as other ecosystems, the decoupling of the invasive pests from their native biocontrol agents is often thought to be the main reason for their successful invasions [16]. *Halyomorpha halys* is a secondary pest often controlled by a suite of coevolved natural enemies that suppresses its population in its native environment in Asia [17]. There are numerous studies on egg parasitoids of *H. halys* both in native and newly invaded areas [5,18,19,20,21,22,23,24,25,26]. A total of 16 species of parasitoids were reported to attack *H. halys* in Asia [18]. The samurai wasp, *Trissolcus japonicus* (Ashmead) (Hymenoptera: Scelionidae), has been identified as the most promising agent for biological control in China [19]. It has also been considered as a classical biological control agent for introduction in invaded areas. In fact, adventive populations of *T. japonicus* are already present in the United States [5,20,21,22,23], Switzerland [24], Italy [25] and Canada [26]. However, the natural enemies of the nymphs and adults are poorly known and could be a complementary component of egg parasitoids during biological control.

Tachinids, such as *Pentatomophaga latifascia* (Villeneuve) (Diptera: Tachinidae), play an important role in shaping the ecological communities of insects [27]. The larvae of tachinids are koinobiont parasitoids, which means that their hosts continue feeding and growing while the parasitoids develop internally [28]. Larvae feed on the hemolymph and nonvital tissues, eventually transitioning to the vital organs. Upon reaching the end of their larval stage, they emerge from the host to pupate, killing the host in the process of their development [27,29]. Their hosts are normally adults, and occasionally, late-instar nymphs [30,31,32]. Tachinids are solitary endoparasites that can lay hundreds of eggs throughout their lifespan [33]. After hatching, tachinid larvae penetrate the host’s body to feed until reaching the last larval instar, when they then emerge to pupate in the soil. *Pentatomophaga latifascia* adults feed on nectar sources from plants and were observed to be active in May, August and September in Beijing, China [34]. *Pentatomophaga latifascia* is distributed in China, Japan, Korea, Russia, India, and Malaysia [34].

Some tachinid species have been found to be natural enemies of pests, including stink bugs. For instance, *Trichopoda pennipes* (Fab.) (Diptera: Tachinidae) parasitizes several heteropteran hosts [35], most commonly *Nezara viridula* (L.) (Pentatomidae) [33], and *H. halys* from North America [36]. For this study, we hypothesized that some tachinid flies could parasitize *H. halys* adults in China. The overwintering populations of *H. halys* were collected and checked before they emerged in spring, indicating that *P. latifascia* was a parasitoid of *H. halys*. The parasitism and morphological characteristics of *P. latifascia* were further examined to provide valuable information that may be potentially used for biological control of *H. halys* targeting adults and nymphs.

## 2. Materials and Methods

*Halyomorpha halys* adults were collected at three different locations close to mountains, from 21 to 24 October 2019. The first location was a botanical garden (40°00′21″ N; 116°11′55″ E), near Fragrant Hill, where we checked old empty beehives surrounding the research building. The second location was in Lengquan village (40°02″06″ N; 116°12′41″ E), near Baiwang Mountain, where we observed various items in an artificial storage facility on a farm. The third location was near Yangtai Mountain (40°04′13″ N; 116°04′55″ E), where some abandoned wooden doors and chairs were examined for *H. halys* adults in a simple storage shelter. All adult *H. halys* found were placed into containers, separated by location, and brought to the laboratory. The body surfaces of *H. halys* specimens were observed under a microscope (Olympus SZ2-ILST) (Olympus Corporation, Olympus Co. Ltd., Tokyo, Japan) and individuals carrying tachinid eggs were recorded and reared according to each location. The *H. halys* adults were fed green beans (*Phaseolus vulgaris* L.) and corn ears (*Zea mays* L.) and maintained in nylon mesh-screened rearing cages (60 × 60 × 60 cm), at 25 ± 1 °C, 65 ± 5% RH, and a 16:8 h L:D photoperiod until tachinid fly eclosion.

The tachinid flies were identified by Guoyue Yu (Institute of Plant and Environment Protection, Beijing Academy of Agriculture and Forestry Sciences). Adult flies are characterized by two golden horizontal bands on the thorax, one extending to the lateral margin. The abdomen is reddish brown and each abdominal segment shows a black or brown posterior margin [34]. The samples were stored in the MARA−CABI joint laboratory, Beijing, China. Each individual *H. halys* carrying eggs of *P. latifascia* was recorded as parasitized, and the parasitism rate was calculated by the number of parasitized *H. halys* divided by the total number of stink bugs for each location.

## 3. Results

### 3.1. Parasitism of H. halys Caused by P. latifascia

We collected 2011 adults of *H. halys* in the botanical garden, 442 adults in Lengquan village, and 148 adults in Yangtai Mountain, and 60, 7 and 4 adults were found carrying eggs of *P. latifascia*, with parasitism of 2.98%, 1.58% and 2.70%, respectively (Table 1). The average parasitism was 2.42 ± 0.43%. Only one *H. halys* adult was found with two *P. latifascia* eggs, the other individuals had only one tachinid fly egg. At least 31 pupae and 19 adults of *P. latifascia* successfully developed. The real number of pupae and flies was likely higher than this, but we were unable to properly sustain the *H. halys* specimens due to COVID-19.

### 3.2. Morphology Characteristics of P. latifascia

The egg of *P. latifascia* was oval-shaped and white or gray in color. It was always deposited on the pronotum or scutellum of *H. halys* (Figure 1A). The mean length and width of the eggs were 0.50 ± 0.01 mm and 0.34 ± 0.01 mm, respectively (Table 2).

The larva of *P. latifascia* penetrated the body of *H. halys* and fed internally on the host’s tissues. Upon reaching the end of its larval stage, it emerged from the *H. halys* body and pupated externally (Figure 1B). We did not obtain enough tachinid larvae to measure their size when they were inside the body of the *H. halys*.

The pupae of *P. latifascia* were long, elliptical and dark reddish brown (Figure 1C). Its mean length and width were 6.50 ± 0.08 mm and 2.79 ± 0.05 mm, respectively (Table 2). The development time from when specimens were collected to the appearance of the pupa was 21.65 ± 0.65 days.

The average length of adult *P. latifascia* (from the head to end of the abdomen) was 7.12 ± 0.07 mm, and the width (the widest part of the abdomen) was 2.40 ± 0.04 mm (Table 2). The front side was covered with golden villi. The thorax had two golden horizontal bands, one of which extended to the lateral margin. The forewing was infuscate. Tarsae were black. The coxa and trochanter were reddish brown. The abdomen was reddish brown, and each abdominal segment had a large or small black or brown posterior margin (Figure 1D).

## 4. Discussion

*Pentatomophaga latifascia* was found attacking and killing *H. halys* adults in our survey, and the average parasitism level was 2.42% on overwintering populations in Beijing. This discovery adds important knowledge about the natural enemy community composition of *H. halys* in its native range and may have positive implications for increasing the biological control in newly invaded areas. Several tachinid fly species have been reported as parasitic on adults, or both adults and nymphs of *H. halys* in newly invaded countries [30,31,32]. Aldrich et al. studied *T. pennipes*, *Euclytia flava* (Townsend) (Tachinidae: Phasiinae), *Gymnosoma par* (Walker) (Tachinidae: Phasiinae) and *Euthera tentatrix* (Loew) (Tachinidae: Dexiinae) on several Pentatomidae using choice tests in the laboratory, and found that all of the tested tachinid flies could lay eggs on *H. halys*. *Trichopoda pennipes* and *E. flava* were further confirmed as parasitoids of *H. halys* in the field, and the parasitism level was approximately 2% in Allentown, PA, USA [30]. Joshi et al. examined the extent of *H. halys* parasitism by the native parasitoid *T. pennipes* in Pennsylvania, and their results showed that the overall parasitism level was 2.38% [31]. Duncan observed that both adults and nymphs of *H. halys* were found bearing the eggs of *Gymnoclytia occidua* (Walker) (Diptera: Tachinidae) in the field [32]. No nymphs of *H. halys* were collected from our survey, because we only sampled the overwintering adult population, and *H. halys* normally starts migration to overwintering sites in early October in Beijing [37]. However, we found that *P. latifascia* indeed laid eggs on nymphs of *H. halys* in rearing cages under laboratory conditions, indicating that *P. latifascia* has the potential to control nymphs and adults of *H. halys* in the field. Tachinids contribute to parasitism on adults and nymphs of *H. halys* in newly invaded countries and also in its native area [30,31], which may be useful to develop an improved biological control strategy of *H. halys* by enriching the natural enemy community composition.

Further investigation on *P. latifascia* as a potential biocontrol agent is required, as many tachinid species have been used as biological control agents [38,39]. There have also been examples of tachinid parasitoids being artificially reared and used as biological control agents for stink bugs. Pickett et al. released *T. pennipes* to control *Anasa tristis* (Degeer) (Hemiptera: Coreidae), and the realized parasitism rate varied from 1.3% to 92.2% at an organic farm in California [36]. Coombs and Sands utilized sugar cubes and moistened cloth pads as food sources, and provided adult bugs as hosts of *Trichopoda giacomellii* (Blanchard) (Diptera: Tachinidae) to maintain the tachinid population in the laboratory, and then directly released the *T. giacomellii* adults for biological control of *N. viridula* on the castor oil plant (*Ricinus communis* L.) in Australia. The parasitism of *N. viridula* adults by *T. giacomellii* ranged from 9% to 72%, including 42% of adults during diapause [40]. Based on these successful cases of biological control of Hemiptera by tachinid flies, it may be possible to mass rear and release *P. latifascia* to control *H. halys*.

Using tachinids as biological control agents also benefits the landscape, because pollinators are vital to creating and maintaining habitats and ecosystems. *Pentatomophaga latifascia* adults use nectar sources from flowering plants and also serve as pollinators of these plants, benefiting the ecological environment. *Pentatomophaga latifascia* functioning as a pollinator has been observed on *Solidago rugosa* (Mill.) (Asterales: Asteraceae) in September in Beijing [34]. Furthermore, Asteraceae and Apiaceae are known to support populations of tachinids in agricultural areas [41]. It has been confirmed that greater parasitism of stink bugs by adult parasitoids (including Tachinidae) was observed where usages of insecticides were reduced [42]. *Pentatomophaga latifascia* populations should be conserved and utilized as part of a community of indigenous natural enemies and pollinators in the environment.

## 5. Conclusions

*Pentatomophaga latifascia* females lay eggs on the surface of *H. halys*, and the hatched larvae penetrate the host body and feed on the hemolymph and nonvital tissues until eventually consuming the vital organs to kill the host. The average parasitism rate of *H. halys* caused by *P. latifascia* was 2.42%, according to our field survey in Beijing, China. This is a new discovery about a parasitoid of adult *H. halys* in its native environment, and has significant potential as a complementary approach to biological control of *H. halys* using egg parasitoids. Furthermore, *P. latifascia* adults feed on nectar sources from host plants and act as pollinators, enhancing ecosystem services to the natural environment. Exploiting *P. latifascia* as a biocontrol agent would not only help suppress the *H. halys* population but also help create a sustainable agricultural ecosystem.

## Figures and Tables

**Figure 1 insects-11-00666-f001:**
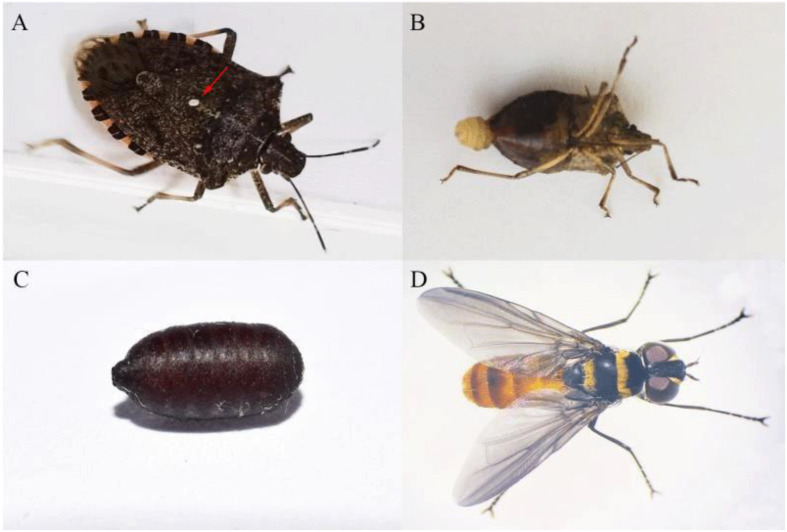
(**A**) Egg of *Pentatomophaga latifascia* on *Halyomorpha halys* body surface (photography by Jinping Zhang); (**B**) Larva of *P. latifascia* emerging from *H. halys* (photography by Juhong Chen); (**C**) Pupa of *P. latifascia* (photography by Guoyue Yu); (**D**) Adult of *P. latifascia* (photography by Wenjing Li).

**Table 1 insects-11-00666-t001:** Parasitism of *Halyomorpha halys* adults by *Pentatomophaga latifascia* at each overwintering location in Beijing.

Numbers	Botanic Garden	Lengquan Village	Yangtai Mountain
Total number of *H. halys* collected	2011	442	148
Number of *H. halys* which were carrying *P. latifascia* egg(s)	60	7	4
Number of *P. latifascia* pupae	>30	>0	>1
Number of *P. latifascia* adults	>19	>0	>0
Percentage of *H. halys* carrying tachinid egg %	2.98%	1.58%	2.70%

**Table 2 insects-11-00666-t002:** Size parameters of *P. latifascia* at different developmental stages.

Stages	Length (mm)			Width (mm)			Number of Measurements
	Mean	Maximum	Minimum	Mean	Maximum	Minimum	
Egg	0.50 ± 0.01	0.63	0.38	0.34 ± 0.01	0.50	0.25	65
Pupa	6.50 ± 0.08	7.00	5.00	2.79 ± 0.05	3.25	2.00	30
Adult	7.12 ± 0.07	7.60	6.50	2.40 ± 0.04	2.70	2.00	19

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
