# Peer review of "A Newly Reported Parasitoid, Pentatomophaga latifascia (Diptera: Tachinidae), of Adult Halyomorpha halys in Beijing, China"

_insects, 2020, doi:10.3390/insects11100666_

Round 1

Reviewer 1 Report

The paper reports on the first find of BMSM adults parasitized by Pentatomophaga latifascia (Diptera: Tachinidae) in the native range of the pest.  The survey was performed in three locations near Beijing (China) collecting a significant number of overwintering adults and resulted in detecting an average parasitism level of H. halys of about 2.4%.

Despite the low percentage of parasitization, the discovery is noteworthy because it adds an important piece of knowledge about the natural enemies community composition in the H. halys native areas and could have positive implications in integrating the biological control in the new invaded areas. 

In conclusion, I consider the paper sufficiently novel and interesting to be suitable for publication in Insects.

I would like to raise only few points that are not clear to me.

Line 54-55. Why the AA mention only the characteristic of damage variation on kiwifruits cultivars? Please either justify or put it in a wider context.

Line 56. I think that there are better examples than reference 10 to be cited about polyphagia of BMSB.

Line 63. Hitchhiking potential deserves to be cited among the reasons of the quickly spread of BMSB (see for ex. Lara et al 2018. New Zealand Entomologist  41: 12-24)  

Line 84: Some of the “pest” tachinids  or “best” tachinids??

Table 1. No. of H. halys which deposited P. latifascia egg(s) does not sound correct to me, better: No. of H. halys which were carrying P. latifascia egg(s)

Figure 1A. For more clearness, add an arrow to indicate the egg.

Line 175-176 and 196-197. My opinion is that 2-3% of parasitization is not expected to have such a great impact to reduce damage losses and pesticide treatments. I suggest to dim the concept expressed in this sentences by speaking of an integration of the BC resources. According to the literature, egg parasitoids are far more effective than any other natural enemies in controlling BMSB.     

Reviewer 2 Report

The manuscript "A newly reported parasitoid, Pentatomophaga latifascia (Diptera: Tachinidae), of adult Halyomorpha halys in Beijing, China" is a descriptive study that surveyed adults parasitoids of H. halys. Although the manuscript is relevant because there is no publish data on the subject, its current version is not suitable for publication in its current version. My main concerns with the study are related to the writing and lack of information on important aspects of P. latifascia and H. halys biological relationship. Briefly, the writing on the current version of the manuscript is difficult to follow and understand. Therefore, the authors should consider the use of professional English editing service before resubmission of the manuscript to improve clarity and flow of ideas. Additionally, there is no information on the host range of P. latifascia, and the extent to which it would be specie specific in the management of H. halys – which would suggest the potential impact in future biological control programs targeting H. halys. Thus, I recommend the acceptance of the manuscript only after an extensive revision of its current version. Several comments were made on the attached pdf revision that may be helpful to the authors.

Reviewer 3 Report

The MS report important new information on a parasitoid of BMSB. Although this is an important discovery, the organization of the MS can be improved. There are several sentences where it is difficult to understand.

Line 12: Delete “famous.”

Line 13: change to “research.”

Line 16-18: rewrite the sentence

Line 28: attacked by “pesticides.”

Lines 51-82: Not sure, what is the focus of this paragraph. The authors started with damage then got into biological control. I suggest you split the paragraph into two and focus on the topic.

Line 89: it is not clear to the reviewer how the adults at overwintering were infested by the parasitoid. Is the tachinid flies active at overwintering sites, such as man-made structures? Are they (tachinids) oviposited eggs on adults before BMSB went into winter diapause?

Line 107: What are the characters that help to identify this species? Where are the citations for the key used for identification?

Line 138: Which regions of the insect the measurements were taken.

Line 161: Please provide a citation for this sentence.

Lines 168-177: Make sure authors what cropping systems these parasitism examples were obtained.

Round 2

Reviewer 2 Report

The revised version of the manuscript was improved in comparison to the first version, but the authors still need to work on their writing. In addition, I do not understand why the authors mentioned nymphal infestation in the abstract and discussion but did not report it in their results. The paper should be carefully revised before publication. Please see my comments and suggestions in the pdf attached here.

Reviewer 3 Report

The authors addressed most of the changes. Please incorporate the information below into the manuscript.

  1. Please incorporate this important information into the M&M section of the manuscript.  "Response 6: Halyopmorpha halys started to overwinter at early October in Beijing China, and we collected the H. halys overwinter population at 21th October and 24th October 2019, we deduced that tachinids oviposit eggs on adults before H. halys went to overwinter sites, however, we still need to confirm this." into the manuscript. 
  2. Please edit your response and incorporate "The identification characters were from a Chinese book (see the follow picture). Description as : The head side was covered with golden villi, the side face was gray; the chest had two golden horizontal bands, one of which extends to the lateral margin; the forewing was smokey brown, feet black, leg ganglia base and pitch reddish brown, abdomen reddish brown, and each abdominal ganglia had a large or small black/brown posterior margin." into the manuscript. 
